# One Patch Doesn't Fit All: Adaptive Patching for Native-Resolution Multimodal Large Language Models

**Wenzhuo Liu**[1,2]   **Weijie Yin**[3]   **Fei Zhu**[4]   **Shijie Ma**[1,2]   **Haiyang Guo**[1,2]   **Yi Chen**[1,2]
**Xiaohui Li**[1,2]   **Xiao Liang**[3]   **Chao Feng**[3]   **Cheng-Lin Liu**[1,2*]
[1]School of Artificial Intelligence, University of Chinese Academy of Sciences
[2]State Key Laboratory of Multimodal Artificial Intelligence Systems,
 Institute of Automation, Chinese Academy of Sciences
[3]ByteDance Inc    [4]Centre for Artificial Intelligence and Robotics, HKISI-CAS
{liuwenzhuo2020, mashijie2021}@ia.ac.cn, zhfei2018@gmail.com

## Abstract

Real-world visual signals are inherently variable in resolution, and it is natural to endow multimodal large language models (MLLMs) with such native-resolution perception capabilities. In principle, for general and straightforward multimodal understanding, low-resolution images are sufficient. While for images with nuanced details like documents and charts, it is crucial to preserve fine-grained details using high-resolution inputs, as naive resizing inevitably results in information loss. Recent advances employ sequence packing to process images of any resolution and aspect ratios. Despite these efforts, model performance degrades at both low and high resolutions, and high-resolution inputs incur substantial computational costs. We argue that the rigid use of a single patch size is the primary cause: when image resolution or information density varies, fixing patch size is intrinsically suboptimal. To address this issue, we introduce *Adaptive Patching (AdaPatch)*, a simple yet effective strategy that adjusts patch size according to image resolution and information density and could be seamlessly plugged into pre-trained fixed-patch MLLMs without any training efforts. Extensive evaluations demonstrate consistent improvements in native resolution performance without additional training. Besides, we provide a training-based method to further adapt MLLMs with dynamic patch sizes and enhance the performance.

## 1 Introduction

Multimodal large language models (MLLMs) (Liu et al., 2024a; Team et al., 2023) have emerged as a central paradigm for joint visual and linguistic understanding. A common design couples a pretrained vision transformer with a large language model via a lightweight projector. In real-world scenarios, visual inputs exhibit substantial variability in resolution and aspect ratio, ranging from low-resolution thumbnails to high-resolution documents, making robust native resolution capability a key requirement for MLLMs (Guo et al., 2024). Early MLLMs (*e.g.*, the LLaVA (Liu et al., 2023a; 2024b) and InternVL (Chen et al., 2024b) series) typically rely on fixed-resolution vision encoders, resizing images to a single canonical size or partitioning them into tiles prior to encoding. Such preprocessing inevitably alters visual content and can degrade performance on inputs that demand preservation of fine detail or global structure, *e.g.*, charts and diagrams.

Following NaViT (Dehghani et al., 2023), recent works retain images at native resolution and divide them into fixed-size, non-overlapping patches, producing token sequences whose length scales with image size. Multiple sequences are concatenated into a single packed sequence and processed jointly by the encoder using per-sample image masks, a technique known as sequence packing. Several state-of-the-art models, including Qwen2.5-VL (Bai et al., 2025), Ovis2.5 (Lu et al., 2025), and Kimi-VL (Team et al., 2025), implement this paradigm and claim support for any input resolutions.

---

*Corresponding author.

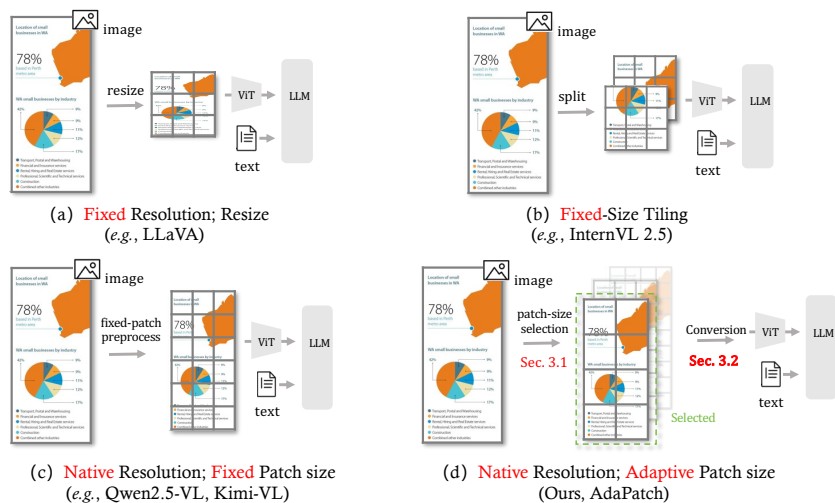

Figure 1: Illustration of strategies for different image resolutions: (a) Fixed resolution: resize to a preset size; (b) Fixed-size tiling: split into uniform tiles; (c) Native resolution: fixed patch size with sequence packing; (d) Our method: adaptively adjust patch size.

In this work, we demonstrate that current designs for any-resolution processing still fail to realize genuine native capability. Through systematic evaluation on multiple benchmarks across a broad spectrum of pixel range, we reveal that model performance fluctuates considerably between resolution bands. It often degrades or becomes unstable at both low and high resolutions. These findings suggest that robust any-resolution understanding remains an open challenge.

Our analysis identifies the inflexibility of a fixed patch size as the principal source of degradation. In low-resolution or information-dense images, a large patch size is too coarse to recover fine-grained cues. In contrast, in very high-resolution or information-sparse images, a small patch size becomes overly local and fails to capture global context. This mismatch constrains the model's receptive granularity and underlies most of the observed performance loss.

To address this limitation, we propose *Adaptive Patching*, a simple drop-in method that computes a lightweight information-density estimate $\rho$ and maps $(\rho, r) \mapsto s$ to determine an appropriate patch size $s$ for an image of native resolution $r$, as illustrated in Fig. 1. We further present a weight-preserving conversion that enables pretrained fixed-patch MLLMs to operate with any patch sizes without additional training, effectively transforming them into adaptive patch models while remaining compatible with sequence packing. Extensive experiments show *Adaptive Patching* improves accuracy and stability across resolutions on multiple benchmarks, while reducing encoder token counts at high resolutions to speed up inference. Our contributions are summarized as follows:

- We conduct a comprehensive evaluation of AnyRes performance by rescaling benchmarks across wide resolution ranges, revealing significant degradation and instability in recent state-of-the-art MLLMs with fixed patch size.

- We uncover that the fixed patch size is the principal architectural cause of such degradation: it induces a representational mismatch across resolutions and information densities.

- We propose *Adaptive Patching (AdaPatch)* that adapts patch size according to the resolution and information density. Our method provides both training-free and training-based alternatives. Extensive experiments validate the improved accuracy and stability across resolutions and reduced computation for high-resolution images.

## 2 ANYRES EVALUATION AND ANALYSIS

### 2.1 BACKGROUND AND NOTATION

We consider a causal MLLM that takes an image $\boldsymbol{x}$ and a text sequence $y_{1:T}$ as input and generates an autoregressive distribution $p(y_t \mid y_{<t}, \boldsymbol{x})$ over the next token, including the following steps:

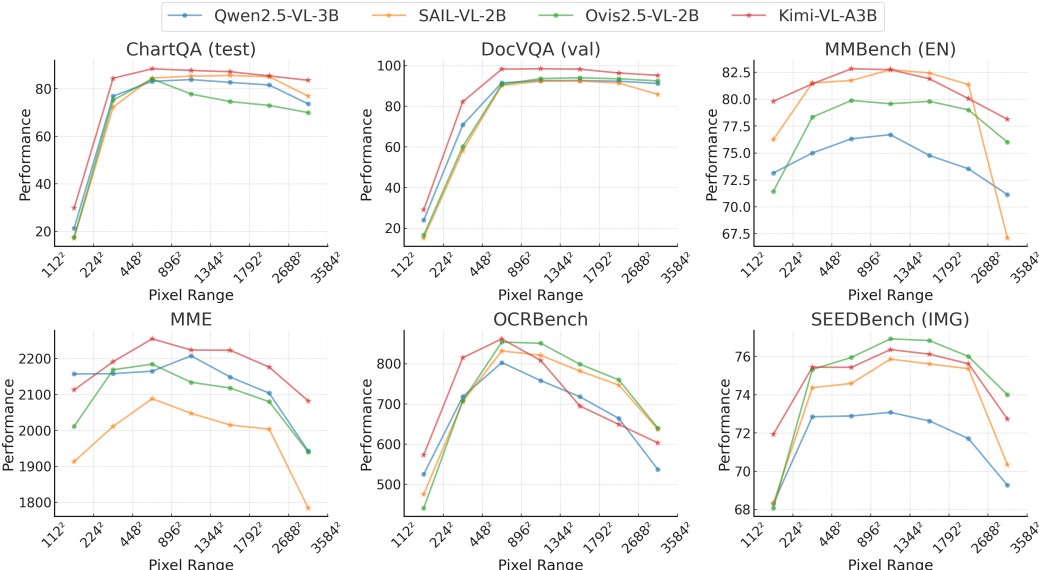

Figure 2: Anyres evaluation of SOTA MLLMs (Qwen2.5-VL, SAIL-VL, Ovis2.5, and Kimi-VL): varying the preprocessing pixel range on a single benchmark causes marked changes in performance, contradicting claims of any-resolution robustness.

**AnyRes preprocessing.** Given an input image $x \in \mathbb{R}^{h \times w \times c}$ with native resolution $r = (h, w)$, the AnyRes preprocessor produces $\tilde{x} \in \mathbb{R}^{\tilde{h} \times \tilde{w} \times c}$ at target resolution $\tilde{r} = (\tilde{h}, \tilde{w})$. The target dimensions $\tilde{h}, \tilde{w}$ are required to be divisible by the patch size $s^1$ and approximate the native size. Formally, $\tilde{r}$ is chosen to minimize:

$$\min_{\tilde{h}, \tilde{w} \in \mathbb{Z}_+, \, \gamma > 0} (\tilde{h}w - \tilde{w}h)^2 + \varepsilon[(\tilde{h} - \gamma h)^2 + (\tilde{w} - \gamma w)^2] + \varepsilon^2(\gamma - 1)^2,$$
$$\text{s.t.} \quad \tilde{h}, \tilde{w} \in s\mathbb{Z}_+, \; P_{\min} \leq \tilde{h}\tilde{w} \leq P_{\max}, \tag{1}$$

where $[P_{\min}, P_{\max}]$ is the valid pixel range, $\gamma$ is an auxiliary scale parameter, and $0 < \varepsilon \ll 1$ is a small tie-breaking weight. Resizing (*e.g.*, bilinear interpolation) is applied independently to each channel and can be regarded as a linear transformation:

$$\tilde{x}_i = \text{resize}_r^{\tilde{r}}(x_i) = B_r^{\tilde{r}} \text{vec}(x_i) \quad (i = 1, \ldots, c), \tag{2}$$

where $\text{vec}(x_i) \in \mathbb{R}^{hw}$ vectorizes channel $i$ and $B_r^{\tilde{r}} \in \mathbb{R}^{\tilde{h}\tilde{w} \times hw}$ is the interpolation matrix. The preprocessed image is $\tilde{x} = [\tilde{x}_1, \ldots, \tilde{x}_c]$.

**Patch embedding.** Divide the image $x \in \mathbb{R}^{h \times w \times c}$ into $n$ non-overlapping $s \times s$ patches ($n = \lfloor h/s \rfloor \lfloor w/s \rfloor$) and embed into vision feature space $\mathbb{R}^{d_v}$. The patch-embedding layer $g_\theta : \mathbb{R}^{h \times w \times c} \to \mathbb{R}^{n \times d_v}$ is a stride-$s$ convolution with kernel $w_\theta \in \mathbb{R}^{s \times s \times c \times d_v}$ and bias $b_\theta \in \mathbb{R}^{d_v}$, i.e. $g_\theta(x) = \text{conv}_\theta(x; s)$, then flattened to an $n \times d_v$ token sequence.

**Vision encoder, projector, and LLM.** From the preprocessed image, patch tokens $Z = g_\theta(\tilde{x})$ are passed to a vision encoder $\mathcal{E}_\phi$ with an attention mask $M$, producing features $V = \{v_j\}_{j=1}^{L_v} = \mathcal{E}_\phi(Z; M)$ (when packing multiple images, $M$ is block diagonal to avoid cross-image attention). A lightweight projector $\Pi_\psi : \mathbb{R}^{d_v} \to \mathbb{R}^{d_\ell}$ maps tokens into the language space, $\hat{V} = \{\hat{v}_j\}_{j=1}^{L_v}$ with $\hat{v}_j = \Pi_\psi(v_j)$. The causal LLM $\mathcal{L}_\xi$ then reads the interleaved sequence $u = [\langle \text{BOS} \rangle, \langle \text{IMG} \rangle, \hat{V}, \langle /\text{IMG} \rangle, \tau(y_{1:T})]$ and the next-token distribution is $p_\xi(y_t \mid y_{<t}, \hat{V}) = \text{Softmax}(\text{Head}(\mathcal{L}_\xi(u_{\leq t})))$, with causal masks enforcing left-to-right text generation.

---

[1]To simplify the analysis, we ignore special-token merging. Under this assumption the factor is equal to the patch size; if special-token merging is applied, the factor is the patch size $\times$ the merge size.

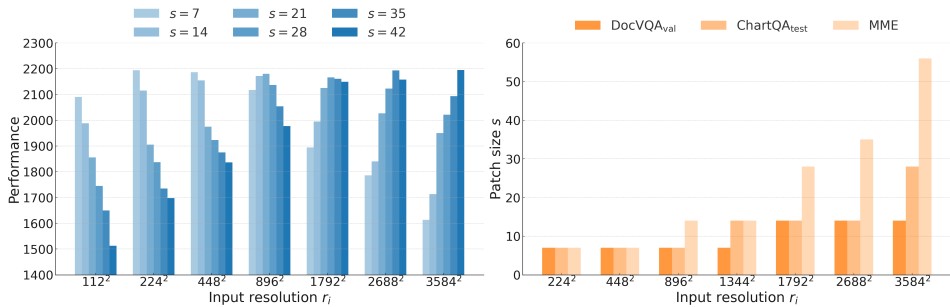

Figure 3: Effect of patch size on AnyRes performance. Left: MME benchmark performance across patch sizes ($s$) and input resolutions ($r$). Right: Relative optimal patch size at the same resolutions for MME, chart, and document images.

## 2.2 ANYRES EVALUATION OF SOTA MLLMS

To explore the ability of handling inputs with various resolutions, for the first time, we evaluate MLLMs from a *pixel-range* perspective: we apply multiple pixel ranges $P \in [P_{\min}, P_{\max}]$ during AnyRes preprocessing and evaluate representative models (Qwen2.5-VL, Ovis2.5, Kimi-VL, and SAIL-VL). Experimental Results in Fig. 2 show that different models (and tasks) exhibit distinct preferred resolution ranges, and the performance drops sharply at both low and high pixel ranges: (1) Low pixel budgets: the degradation is pronounced on information-dense images (*e.g.*, charts, documents) because resampling at low resolutions range $P$ destroys fine-grained signals, making it difficult to extract accurate cues from information-dense images. (2) High pixel budgets: most models decline markedly despite claims of supporting ~3K inputs. We argue that a fixed (relatively small) patch size struggles to capture global context for information-sparse images with high resolution. In conclusion, a fixed patch size fails to cope with images at varying resolutions.

## 2.3 HOW FIXED PATCH SIZE DRIVES ANYRES INSTABILITY

To verify our assumptions and investigate the effect of patch size in Anyres, we convert pretrained Qwen2.5-VL (patch size $s = 14$) to multi-sizes $s \in \{7, 14, 21, 28, 35, 42\}$ (details in Sec. 3.2) and evaluate each configuration on the MME benchmark across a wide range of input sizes. The results in Fig. 3 (Left) reveal a clear, resolution-dependent preference: smaller patches improve performance at low resolutions, while larger patches are preferable at high resolutions.

However, the preferred patch size is not determined by resolution alone but also depends on *information density*. Fig. 3 (Right) compares three image types: documents, charts, and general MME images. For a given absolute resolution, the empirically optimal patch size is smaller for higher information-density images (*e.g.*, $\rho_{\text{document}} > \rho_{\text{chart}} > \rho_{\text{MME}}$, where $\rho$ denotes the information density). Empirically, we observe the following approximate trend: $s^\star \propto r/\rho$ where $s^\star$ is the preferred patch size and $r$ denotes a scalar measure of resolution (*e.g.*, pixels on the shorter image side).

## 3 METHOD: ADAPTIVE PATCHING

Motivated by the observation in Sec. 2 that the optimal patch size is related to native resolution and information density, we propose *Adaptive Patching (AdaPatch)*. Our method quantifies information density and maps it (together with native resolution) to a patch-size estimate. We then extend fixed-patch models to any-patch, enabling per-image patch-size selection. The proposed method is illustrated in Fig. 4 and described below.

### 3.1 ADAPTIVE PATCH SIZE ESTIMATION

**Estimation of Information Density.** In this work, we introduce a simple measure of image information density to guide patch size adaptation. Intuitively, downsampling high-density images leads to larger information loss than downsampling sparse images. To capture this effect, we quantify information density by comparing features extracted from the native images with those from a

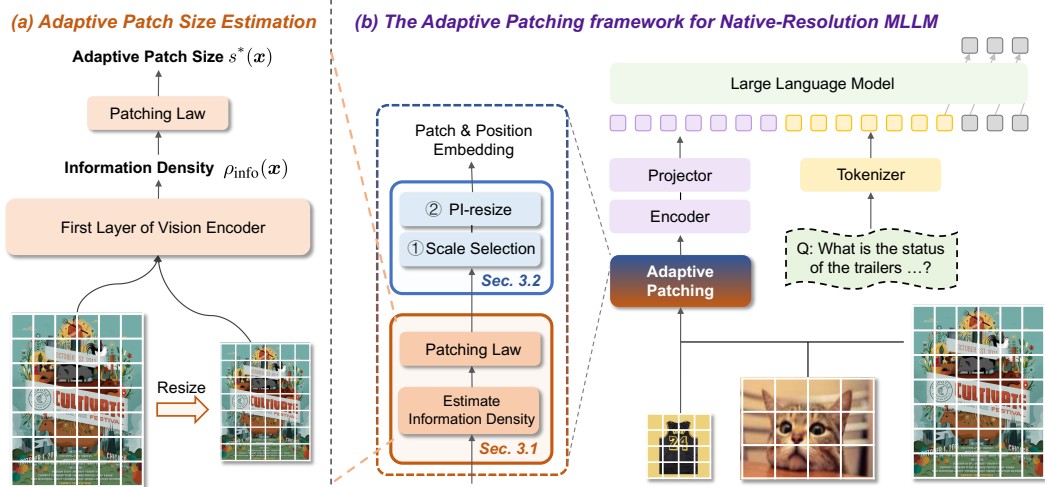

Figure 4: *Patching* is a drop-in method that estimates information density $\rho$ from feature similarity and maps $(\rho, r) \mapsto s$ to choose a patch size. We also convert fixed-patch MLLMs to any-patch models for per-sample adaptation at training or inference.

decreased resolution. Formally, the information density of an image $\boldsymbol{x}$ is defined as

$$\rho_{\mathrm{info}}(\boldsymbol{x}) = 1 - \frac{1}{n} \sum_{i=1}^{n} \frac{\left\langle \mathcal{E}_{\boldsymbol{\phi}}^{[k,i]}(\mathrm{conv}_{\boldsymbol{\theta}}(\tilde{\boldsymbol{x}}; s)), \ \mathcal{E}_{\boldsymbol{\phi}}^{[k,i]}(\mathrm{conv}_{\boldsymbol{\theta}*}(B_r^{r/2} \ \mathrm{vec}(\tilde{\boldsymbol{x}}); s/2)) \right\rangle}{\left\| \mathcal{E}_{\boldsymbol{\phi}}^{[k,i]}(\mathrm{conv}_{\boldsymbol{\theta}}(\tilde{\boldsymbol{x}}; s)) \right\| \cdot \left\| \mathcal{E}_{\boldsymbol{\phi}}^{[k,i]}(\mathrm{conv}_{\boldsymbol{\theta}*}(B_r^{r/2} \ \mathrm{vec}(\tilde{\boldsymbol{x}}); s/2)) \right\|}. \tag{3}$$

Here, $\mathcal{E}_{\boldsymbol{\phi}}^{[k,i]}$ denotes the $i$-th token feature at layer $k$ of the vision encoder, and $\mathrm{conv}_{\boldsymbol{\theta}*}$ is the patch embedding with patch size adjusted to $s/2$. In definition, $\rho(\boldsymbol{x}) \in [0, 1]$ where larger values signify increased downsampling loss and therefore higher information density.

**Adaptive Patching Law.** We adjust the patch size based on the empirical observation in Sec. 2.3 that $s^\star \propto \frac{r}{\rho}$. For simplicity, we model this dependency as a power-law relation and introduce hyperparameters $\alpha > 0$ and $\beta > 0$ to control the relative sensitivity to resolution and information density, respectively. Specifically, the target patch size $s^\star$ for image $\boldsymbol{x}$ is given by

$$s^*(\boldsymbol{x}) = \mathrm{Quantize}\left( \mathrm{clip}\left( \tilde{s} \left( \frac{\kappa(r_{\boldsymbol{x}})}{r_0} \right)^{\alpha} \left( \frac{\tilde{\rho}}{\rho(\boldsymbol{x}) + \varepsilon} \right)^{\beta}, s_{\min}, s_{\max} \right) \right). \tag{4}$$

where $\kappa(\cdot)$ is a scalar measure of resolution (*e.g.*, $\min\{h, w\}$), and $r_0$, $\tilde{\rho}$, and $\tilde{s}$ denote the base resolution, information density, and patch size of the pretrained model, with default values set to 896, 14, and 0.2. The constant $\varepsilon$ is a small positive value for numerical stability. The operator $\mathrm{clip}(\cdot, s_{\min}, s_{\max})$ constrains to specified bounds, while $\mathrm{Quantize}(\cdot)$ maps the result to the nearest value from a predefined discrete set in $\mathbb{Z}_+$.

Eq. 4 reflects two trends: (1) higher resolution favors larger patches, and (2) higher information density favors smaller patches to preserve details. This rule enables per-image patch size selection that balances resolution and density, remaining compatible with existing sequence-packing MLLMs.

## 3.2 CONVERTING FIXED-PATCH MLLMS TO ANY-PATCH

Given a pretrained MLLM with patch embedding $g_{\boldsymbol{\theta}}$ at patch size $s$, our objective is to extend it to any sizes $\{s_i\}_{i=1}^{M}$ while preserving model performance. We introduce two solutions: pseudo-inverse resize (training-free) and multi-scale patch embedding (training-based).

**Pseudo-inverse resize** (training-free) optimizes the patch-embedding layer $\mathrm{conv}_{\boldsymbol{\theta}_i}$ so that token embeddings are consistent across patch sizes $s_i$ and corresponding resolutions $r_i$. The optimization objective is:

$$\{\boldsymbol{\omega}_{\boldsymbol{\theta}_i}, \boldsymbol{b}_{\boldsymbol{\theta}_i}\} := \boldsymbol{\theta}_i = \arg\min_{\boldsymbol{\theta}_i} \ \mathbb{E}_{\boldsymbol{x} \sim \mathcal{X}} \left\| \mathrm{conv}_{\boldsymbol{\theta}}(\boldsymbol{x}, s) - \mathrm{conv}_{\boldsymbol{\theta}_i}\left( B_r^{r_i} \ \mathrm{vec}(\boldsymbol{x}), s_i \right) \right\|_F, \quad i = 1, \dots, M. \tag{5}$$

Table 1: **Evaluation results under native resolution.** Our method (*Adaptive Patching*) is compared with the baseline (fixed patch size = 14), using the same pretrained model. For each model, the larger score between vanilla and our is marked in **bold**.

| Benchmark | Qwen2.5-VL-3B | | SAIL-VL-2B | | Ovis2.5-2B | | Kimi-VL-A3B | |
|---|---|---|---|---|---|---|---|---|
| | *vanilla* | *AdaPatch* | *vanilla* | *AdaPatch* | *vanilla* | *AdaPatch* | *vanilla* | *AdaPatch* |
| **General** | | | | | | | | |
| MME | 2135.90 | **2210.41** | 2026.27 | **2103.96** | 2156.29 | **2208.60** | 2191.92 | **2230.94** |
| MMMU | 49.22 | **51.89** | 44.11 | **45.97** | 46.33 | **49.28** | 50.44 | **51.32** |
| MMStar | 54.27 | **55.42** | 59.87 | **60.68** | **59.47** | 59.22 | 50.93 | **52.86** |
| MMBench$_{EN}$ | 75.15 | **78.49** | 81.58 | **83.35** | 78.79 | **81.32** | 81.58 | **82.50** |
| LLaVABench | **66.30** | 65.34 | 52.50 | **53.53** | 42.00 | **44.81** | **70.00** | 68.70 |
| RealWorldQA | 65.36 | **66.35** | 69.80 | **70.76** | 67.84 | **68.15** | 69.80 | **70.30** |
| SEEDBench$_{IMG}$ | 72.91 | **75.57** | 74.40 | **75.91** | 75.37 | **76.36** | 75.57 | **77.77** |
| AI2D$_{test}$ | 80.31 | **81.16** | 82.32 | **82.67** | **84.55** | 84.26 | 80.47 | **80.87** |
| HallusionBench | 48.73 | **50.80** | 51.32 | **53.44** | 50.70 | **52.35** | 48.26 | **50.52** |
| **Domain** | | | | | | | | |
| ChartQA$_{test}$ | 82.92 | **83.40** | 84.08 | **85.98** | 83.60 | **84.11** | 88.40 | **89.17** |
| TextVQA$_{val}$ | 78.68 | **79.53** | 79.92 | **80.02** | 79.91 | **80.76** | 89.82 | **89.96** |
| OCRBench | 821 | **845** | 783 | **855** | 706 | **814** | 865 | **890** |
| DocVQA$_{val}$ | 92.31 | **92.62** | 91.48 | **92.89** | **93.90** | 93.22 | 96.45 | **98.33** |
| POPE | 86.39 | **86.88** | **86.10** | 86.09 | 87.81 | **88.65** | 85.75 | **86.43** |
| InfoVQA$_{val}$ | 74.65 | **75.29** | 70.61 | **71.62** | 76.48 | **77.21** | 84.92 | **85.86** |
| MathVista$_{MINI}$ | 60.30 | **62.37** | 67.40 | **69.99** | 63.10 | **66.90** | 65.00 | **67.04** |

The closed-form solution is $\boldsymbol{\omega}_{\theta_i} = B_i(B_i^\top B_i)^{-1}\boldsymbol{\omega}_\theta = (B_i^\top)^+\omega_\theta$, where $B_i = B_{s_2^2}^{s_i^2}$ and $(\cdot)^+$ denote the Moore–Penrose pseudoinverse. For upsampling ($s_i > s$), inner products are preserved exactly, *i.e.*, $\langle B_i\boldsymbol{x}, (B_i^\top)^+\boldsymbol{\omega}_\theta \rangle = \langle \boldsymbol{x}, \boldsymbol{\omega}_\theta \rangle$; for downsampling ($s_i < s$), the pseudoinverse yields the optimal approximation. We introduce PI-resize as a weight-preserving mapping from $s$ to $s_i$:

$$\text{PI-resize}_s^{s_i}(\boldsymbol{w}) = (B_i^T)^+\text{vec}(\boldsymbol{w}). \tag{6}$$

**Multi-scale patch embedding** (training-based) allocate independent parameter $\{\boldsymbol{\theta}_i\}_{i=1}^M$ for patch size $s_i$ and jointly train MLLM with embedding layers end-to-end. During training, the patch size $s_i$ is sampled adaptively, and we minimize the task loss

$$\min_{\{\boldsymbol{\theta}_i\}, \boldsymbol{\phi}, \boldsymbol{\psi}, \boldsymbol{\xi}} \mathbb{E}_{(\boldsymbol{x}, y_{1:T}) \sim \mathcal{X}, s_i} \mathcal{J}\Big(\mathcal{L}_{\boldsymbol{\xi}}\big([\langle\text{BOS}\rangle, \langle\text{IMG}\rangle, \hat{\boldsymbol{V}}_i, \langle/\text{IMG}\rangle, \tau(y_{1:T})]\big), y_{1:T}\Big), \tag{7}$$

where $\mathcal{J}$ is the task loss and $\boldsymbol{\theta}_i$ is initialized via PI-resize$_s^{s_i}(\cdot)$. In summary, PI-resize converts fixed-patch to any-patch models at inference, while MSPE trains embedding weights for multiple patch sizes, both supporting *Adaptive Patching* (Sec. 3.1) without altering the backbone.

## 4 EXPERIMENTS

**Evaluation Setup.** We evaluate on diverse public benchmarks, including comprehensive benchmarks (MME (Fu et al., 2023), MMMU (Yue et al., 2024), AI2D (Kembhavi et al., 2016), MM-Bench (Liu et al., 2023b), LLaVABench (Liu et al., 2023a), RealWorldQA, InfoVQA (Mathew et al., 2022), SEEDBench (Li et al., 2023a), MMStar (Chen et al., 2024a)) and domain-specific benchmarks (ChartQA (Masry et al., 2022), TextVQA (Singh et al., 2019), OCRBench (Liu et al., 2024d), DocVQA (Mathew et al., 2021), POPE (Li et al., 2023b), HallusionBench (Guan et al., 2024), ScienceQA (Lu et al., 2022)). Experiments are conducted on four representative native-resolution MLLMs: Qwen2.5-VL, SAIL-VL, Ovis2.5, and Kimi-VL. We exclude fixed-resolution or cropped-input models (*e.g.*, LLaVA, InternVL) to focus on native-resolution architectures.

**Implementation Details.** All experiments are conducted on 8×A100 GPUs with 80GB memory. We evaluate the official checkpoints of Qwen2.5-VL, SAIL-VL, Ovis2.5, and Kimi-VL using VLMEvalKit (Duan et al., 2024). Candidate patch sizes are set to integers within $[6, 56]$. The information density is estimated from the $i = 0$ layer of the vision encoder with default hyperparameters $(\alpha, \beta) = (0.5, 0.3)$. For MSPE, we train with AdamW (learning rate 1e−5, weight decay 0.01). Inference is performed with a maximum generation length of 512 tokens and temperature 0.

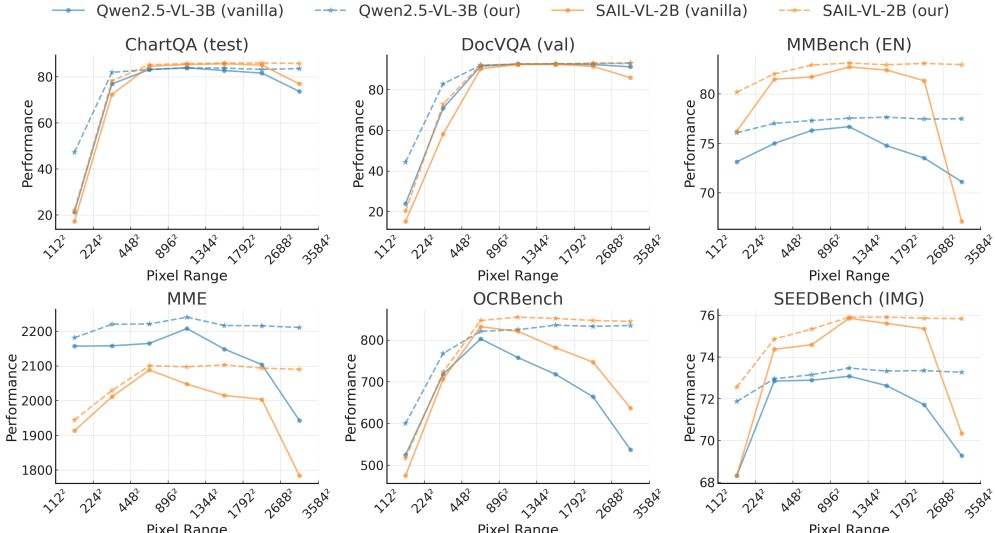

Figure 5: Evaluation results across different pixel ranges of preprocessing pipeline. All MLLMs exhibit notable performance degradation at low and high resolutions, whereas *Adaptive Patching* achieves substantially more stable performance.

Table 2: **Evaluation of *Adaptive Patching* under supervised finetuning.** We evaluate the same pretrained backbone (SAIL-VL-2B) after incremental fine-tuning on LLaVA1.5-665K and LLaVA1.6-779K. Patch sizes are adaptively chosen in our method, while the baseline uses a fixed size of 14.

| Dataset | ChartQA$_{test}$ | DocVQA$_{val}$ | MMBench$_{EN}$ | MME | SEEDBench$_{IMG}$ | TextVQA$_{val}$ | OCRBench |
|---|---|---|---|---|---|---|---|
| LLava1.5-665K | 47.96 | 69.65 | 70.12 | 1821.49 | 71.68 | 68.86 | 418 |
| + *AdaPatch* (Ours) | 54.84 | 70.66 | 71.43 | 1854.14 | 72.24 | 70.34 | 535 |
| LLava1.6-779K | 75.16 | 85.90 | 70.59 | 1790.26 | 71.44 | 72.60 | 466 |
| + *AdaPatch* (Ours) | **77.53** | **87.00** | **71.89** | **1837.25** | **72.04** | **73.59** | **579** |

## 4.1 MAIN RESULTS

**Comparison with State-of-the-Art on Native Resolution.** Table 1 summarizes results across all public benchmarks. *Adaptive Patching* consistently improves performance over the fixed-patch setting across all evaluated MLLMs. The gains are particularly pronounced on benchmarks that involve heterogeneous image resolutions and varying information densities. For instance, OCRBench, which contains images ranging from low resolution ($56 \times 56$) to high resolution ($1344 \times 1344$). Similarly, MMBench and MME, which cover diverse image types and complex information layouts, also benefit notably. On most general benchmarks, our method achieves stable and consistent improvements without introducing additional training overhead. These results confirm that *Adaptive Patching* is especially effective when input resolution differences are large or when the information density varies significantly, thereby enhancing the robustness and adaptability of MLLMs compared to the fixed patch-size baseline.

**Comparison with State-of-the-Art on Pixel-range.** To assess robustness with respect to input resolution, we evaluate models across pixel ranges from $112\times112$ to $3584\times3584$. As shown in Fig. 5, recent MLLMs exhibit strong resolution preferences: performance deteriorates at both low and high pixel budgets, with different models peaking at distinct ranges. In contrast, *Adaptive Patching* markedly mitigates such instability. By dynamically adjusting patch sizes, our method preserves fine-grained detail under limited budgets while maintaining global context at larger scales. Consequently, performance remains substantially more stable across the entire spectrum. These results indicate that resolution sensitivity in current MLLMs largely stems from fixed patching, and *Adaptive Patching* is an effective solution.

**Comparison with SAIL-VL-2B on Supervised Finetuning.** Training-based *Adaptive Patching* is realized by modifying the patch embedding layer (Eq. 7) and initializing multi-scale patch sizes with PI-resize (Eq. 6), where the default set is $\{s_i\}_{i=1}^M = \{8, 12, 14, 16, 24, 28\}$. Under limited computational resources, we incrementally finetuned the SAIL-VL-2B model on the LLaVA1.5-

Table 3: **Performance comparison across different model scales.** The table reports evaluation results of *Adaptive Patching* on Qwen2.5-VL (3B, 7B, 34B) and Ovis2.5 (2B, 9B).

| Model Size | SEEDBench | DocVQA | MMBench | MME | OCRBench | MMMU | TextVQA | Hallusion | LLaVABench | ChartQA |
|---|---|---|---|---|---|---|---|---|---|---|
| | | | | *Qwen2.5-VL* | | | | | | |
| 3B | 72.91 | 92.31 | 75.15 | 2135.90 | 821.00 | 54.27 | 78.68 | 48.73 | **66.30** | 82.92 |
| *+ our method* | **75.57** | **92.62** | **78.49** | **2210.41** | **845.00** | **55.42** | **79.53** | **50.80** | 65.34 | **83.40** |
| 7B | 76.15 | 94.50 | 81.89 | 2313.17 | 863.00 | 62.13 | 84.62 | 54.73 | 74.70 | 85.68 |
| *+ our method* | **76.50** | **94.80** | **82.04** | **2345.96** | **897.00** | **63.55** | **85.38** | **55.91** | **76.79** | **86.42** |
| 32B | 76.47 | 92.47 | 85.84 | 2433.57 | 839.00 | 66.07 | 77.76 | 57.21 | **77.50** | 72.16 |
| *+ our method* | **76.92** | **92.68** | **86.56** | **2460.71** | **859.00** | **66.18** | **79.61** | **58.22** | 77.36 | **74.78** |
| | | | | *Ovis2.5* | | | | | | |
| 2B | 75.37 | **93.90** | 78.79 | 2156.29 | 706.00 | **59.47** | 79.91 | 50.70 | 42.00 | 83.60 |
| *+ our method* | **76.36** | 93.22 | **81.32** | **2208.60** | **814.00** | 59.22 | **80.76** | **52.35** | **44.81** | **84.11** |
| 9B | 77.12 | 95.08 | 84.67 | 2353.73 | 722.00 | 67.00 | 81.85 | 55.96 | **78.30** | 85.92 |
| *+ our method* | **77.19** | **95.19** | **85.51** | **2390.41** | **843.00** | **67.31** | **82.09** | **56.81** | 77.45 | **86.73** |

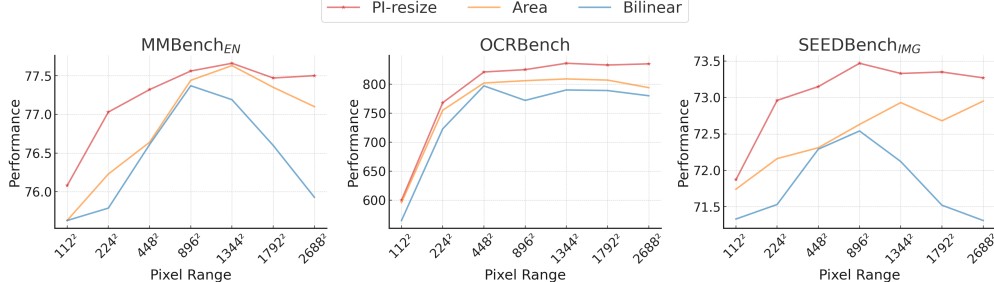

Figure 6: Comparison of different resizing methods on Qwen2.5-VL-3B. We evaluate Bilinear, PI-resize, and Area for adjusting the embedding layer weights, where PI-resize performs the best.

665K and LLaVA1.6-779K datasets, which inevitably leads to some degradation compared to the original pretrained model. During supervised finetuning, the patch size is adaptively selected per image, while the remainder of the training pipeline is kept unchanged. As shown in Table 2, *Adaptive Patching* yields consistently higher performance across datasets at their native resolutions compared to the fixed patch size baseline ($s = 14$). These results indicate that dynamically adjusting patch sizes during training can provide more benefits.

## 4.2 ABLATION STUDY AND ANALYSIS

**Impact of Model Scale.** We examine whether the advantages of *Adaptive Patching* scale with model size by evaluating Qwen2.5-VL (3B, 7B, 34B) and Ovis2.5 (2B, 9B). As shown in Table 3, consistent gains are observed across all scales, indicating that the benefits stem from mitigating the rigidity of fixed patch sizes rather than from model capacity. The improvements are particularly pronounced on OCRBench and MME, consistent with the findings in Sec. 4.1 that tasks involving diverse image resolutions and information densities benefit most.

**Impact of Resizing Methods.** We investigate the effect of alternative interpolation strategies (bilinear, PI-resize, and area) for adapting pretrained models with fixed patch sizes to different target patch resolutions. Using Qwen2.5-VL-3B as a case study, we compare these resizing methods in conjunction with *Adaptive Patching*. As shown in Fig. 6, PI-resize consistently outperforms other approaches, demonstrating its ability to effectively realign the patch embedding layer without additional training. In contrast, bilinear and area interpolation lead to notable performance degradation.

**Impact of Hyperparameters $\alpha$ and $\beta$.** As shown in Fig. 7a, we evaluate different values of the two hyperparameters, which control sensitivity to resolution and information density when adaptively scaling the base patch size ($s = 14$). The results indicate that $\alpha$ has a stronger influence, with larger values leading to notable performance degradation. By default, we set $\alpha = 0.5$ and $\beta = 0.3$.

**Comparison with Resizing Images.** As shown in Table 4, *Adaptive Patching* consistently outperforms direct image resizing. Although resizing brings gains on benchmarks such as OCRBench, it substantially degrades performance on others (*e.g.*, DocVQA and MME). This highlights the limited robustness of resolution adjustment and underscores the importance of preserving native resolution, which enables the model to directly process heterogeneous visual regions without distortion.

Table 4: **Comparison of image resizing and *Adaptive Patching*.** Images are resized to a fixed pixel range [896, 1344] and compared with *Adaptive Patching* on Qwen2.5-VL and SAIL-VL.

| Dataset | ChartQA$_{test}$ | DocVQA$_{val}$ | MMBench$_{EN}$ | MME | SEEDBench$_{IMG}$ | TextVQA$_{val}$ | OCRBench |
|---|---|---|---|---|---|---|---|
| *Qwen2.5-VL-3B* | | | | | | | |
| IMG-resize | 83.16 | 91.53 | 76.32 | 2165.01 | 72.89 | 78.48 | 803.00 |
| Adaptive Patching | **83.40** | **92.62** | **78.49** | **2210.41** | **75.57** | **79.53** | **845.00** |
| *SAIL-VL-2B* | | | | | | | |
| IMG-resize | 84.56 | 90.24 | 81.73 | 2007.54 | 74.59 | 79.43 | 822.00 |
| Adaptive Patching | **85.98** | **92.89** | **83.35** | **2103.96** | **75.91** | **80.02** | **855.00** |

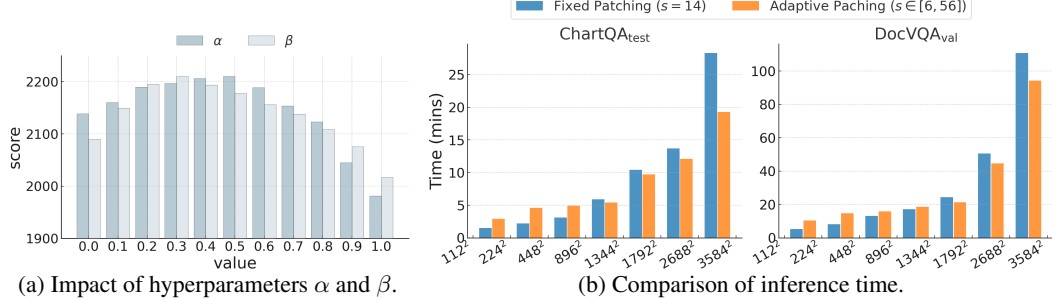

(a) Impact of hyperparameters $\alpha$ and $\beta$.    (b) Comparison of inference time.

Figure 7: Overall impact of hyperparameters and inference efficiency.

**Comparison on Inference Time.** *Adaptive Patching* introduces additional computational overhead from two sources: (1) a lightweight information-density estimation, and (2) variation in the sequence length of image tokens, which is the dominant factor. As shown in Fig. 7b, we compare inference cost against fixed patch sizes. While our approach introduces higher cost at low resolutions, it achieves substantial speedups at high resolutions.

## 5 RELATED WORK

Early works on any-resolution aimed to improve classification accuracy or reduce computation. For example, Learn-to-Resize (Talebi & Milanfar, 2021) replaced fixed interpolation with a learnable resizing layer; Resolution Adaptive Networks (Yang et al., 2020) and Dynamic Resolution Networks (Zhu et al., 2021) routed inputs to different sub-networks or resolutions depending on task difficulty. However, these methods are closely tied to CNN architectures and are difficult to transfer to vision–language systems.

For ViT-based model, FlexiViT (Beyer et al., 2023) resizes patch-embedding weights to trade off accuracy and computational cost. Pix2struct (Lee et al., 2023) and NaViT (Dehghani et al., 2023) preserve native resolution while using a fixed patch size, producing variable-length token sequences and using sequence packing to handle different input sizes. Liu et al. (2024c) extends FlexiViT by learning resized embeddings jointly with the model, which improves cross-resolution performance. Recent MLLMs (*e.g.*, Qwen2.5-VL (Bai et al., 2025)) follow NaViT to process images at native resolution, but a fixed patch size degrades performance at resolution extremes. We propose *Adaptive Patching*, which selects a per-image patch size from the native resolution and an information-density estimate to improve accuracy and efficiency.

## 6 CONCLUSION

We revisit the widely claimed "any-resolution" capability of recent MLLMs and show that their performance is in fact highly sensitive to input resolution, largely due to the rigidity of fixed patch size. To address this, we introduce *Adaptive Patching*, a lightweight and training-free method that adjusts patch size according to image resolution and information density, and converts fixed-patch MLLMs into any-patch models via PI-resize or MSPE. Extensive experiments on a broad suite of benchmarks demonstrate that adaptive patching improves both accuracy and stability across resolutions while reducing computation at high resolutions. Our study highlights patch size as a key bottleneck

for vision–language modeling and provides a simple, general, and effective solution toward genuine native resolution robustness.

**Limitations and Future Work.** In this work, We address the native resolution issue from the perspective of visual processing before LLMs. Future work will incorporate more high-quality multi-resolution data and explore settings with evolving knowledge and continual updates (Guo et al., 2025b;a). We also plan to combine our approach with knowledge-oriented augmentations (Jiang et al., 2025b;a) and efficient vision token processing such as recoverable compression (Chen et al., 2025). Finally, our method may benefit thinking-with-images models like (OpenAI, 2025).

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

# A APPENDIX

## A.1 VALIDATION OF THE INFORMATION-DENSITY FORMULATION

We provide additional analyses to validate the proposed information-density formulation. Representative visualizations are shown in Figure 8 and Figure 9.

**Layer-wise consistency.** Information-density maps computed from different ViT layers exhibit highly similar spatial patterns. Maps from layer 0 closely match those from deeper layers, indicating that the formulation is stable across the feature hierarchy.

**Comparison with statistical measures.** We also compare information-density formulation with two statistical pixel-level measures: gradient entropy, computed from the histogram of Sobel gradient magnitudes, and Laplacian variance, computed from the variance of the Laplacian response. As shown in Figure 9, these traditional measures primarily capture edges and fine-grained textures, whereas the information density defined in Eq. 3 better identifies regions that are semantically meaningful in the feature space.

## A.2 ETHICS STATEMENT

This work adheres to the ICLR Code of Ethics. No human subjects or animal experimentation were involved. All datasets used, including MME, MMMU, AI2D, MMBench, LLaVABench, RealWorldQA, InfoVQA, SEEDBench, MMStar, ChartQA, TextVQA, OCRBench, DocVQA, POPE, HallusionBench, and ScienceQA, are publicly available and widely adopted benchmarks in the research community, ensuring no violation of privacy. We have taken care to avoid biases or discriminatory outcomes in our research process. No personally identifiable information was used, and no experiments were conducted that could raise privacy or security concerns. We are committed to maintaining transparency and integrity throughout the research process.

## A.3 REPRODUCIBILITY STATEMENT

We have taken extensive measures to ensure reproducibility of our results. The experimental setup, including evaluation procedures, model configurations, and hardware details, is described in detail

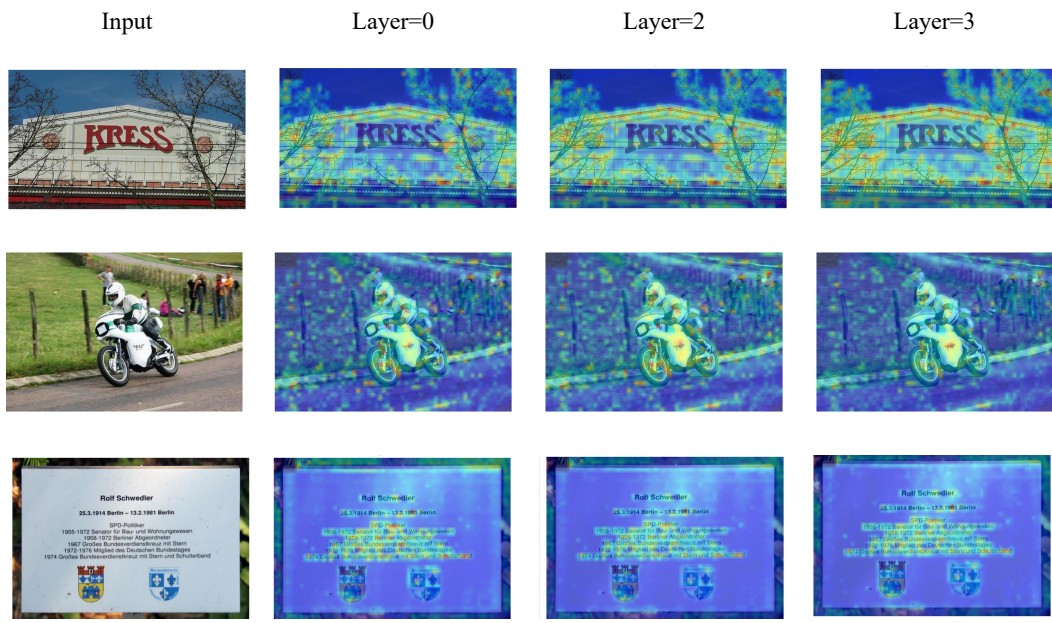

Figure 8: Information-density maps computed across different ViT layers.

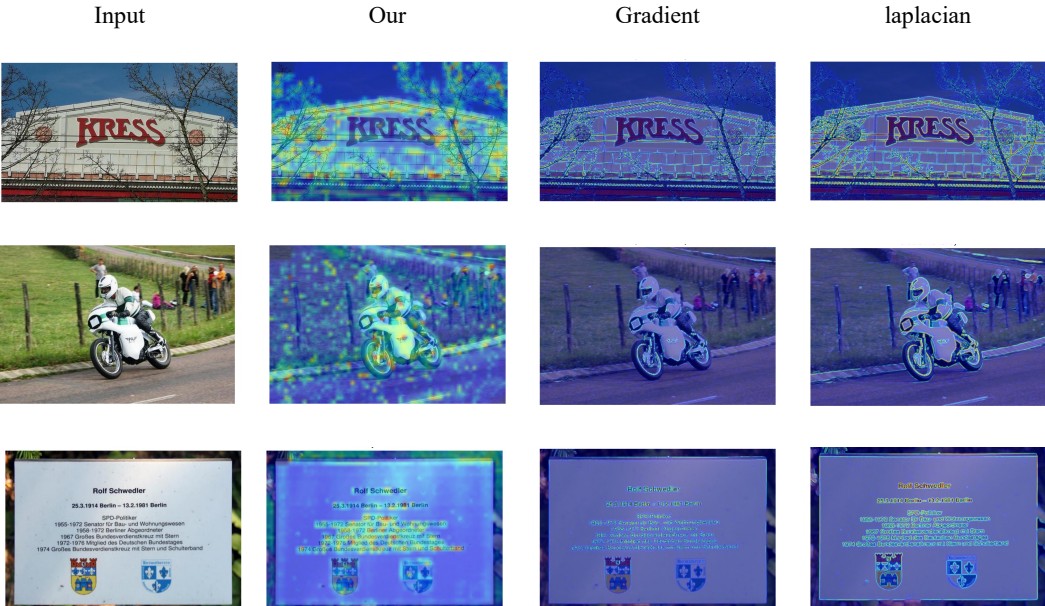

Figure 9: Comparison between the information-density formulation and statistical measures.

in the paper. We evaluate the official checkpoints of Qwen2.5-VL, SAIL-VL, Ovis2.5, and Kimi-VL using open-source code (VLMEvalKit), ensuring consistent and verifiable results. These measures enable other researchers to reproduce our work and build upon it to further advance the field.

## A.4 LARGE LANGUAGE MODELS USAGE

LLMs were used only to assist with writing and polishing the manuscript. They supported tasks such as sentence rephrasing, grammar checking, and improving readability, but were not involved in the ideation, methodology, experimental design, or data analysis. All research concepts, analyses, and conclusions were developed by the authors. The use of LLMs was limited to enhancing linguistic quality, without affecting the scientific content. The authors take full responsibility for

the manuscript and have ensured that the use of LLMs complies with ethical standards and avoids plagiarism or scientific misconduct.

