# OpenReview forum: "One Patch Doesn’t Fit All: Adaptive Patching for Native-Resolution Multimodal Large Language Models"
_ICLR.cc/2026/Conference — ICLR 2026 Poster_

### Official Review · Reviewer_JDNT · 2025-10-19

**Soundness:** 3
**Presentation:** 2
**Contribution:** 3
**Rating:** 4
**Confidence:** 4

**Summary:**

This paper focuses on the native resolution issues of MLLMs, and propose an adaptive patching (AdaPatch) method to achieve the optimal patching and visual embedding of input images. The experiments on a set of experiments show the performance gains obtained by AdaPatch for a set of MLLMs.

**Strengths:**

1. The motivation is clear and plausible. This paper shows that the existing any-res. solution are not optimal due the use of fixed patching. Also, the authors also provide experimental analyses to support their intuitions.

2. The effectiveness of the proposed methods are validated on a set of recent MLLMs. As shown in Tab.1, the proposed AdaPatch can bring performance gains to the compared MLLMs.

**Weaknesses:**

1. The method description is vague. The method section introduces various concepts,  but the most important method procedure lacks a clear and intuitive description. For example, how AdaPatch chooses the optimal patch sizes based on the resolution and information density? Subjectively speaking, this paper over-packages theoretical descriptions but lacks explanation of specific details and procedures.

2. More ablation are required. Although this papers provides a lot of comparisons and charts, it still lacks some critical ablations to show the superiority of the proposed methods. For instance, what about the alternatives to AdaPatch that can also adaptively set patch size? What about the results of using task-specific patch size?

3. Insufficient qualitative analyses. The authors take more efforts to show the resolution inference to MLLMs on different tasks. However, this seems not the key to AdaPatch, since AdaPatch focuses more on the size of patching, as shown in Fig.1. Besides, the authors are required to give more information about the results of AdaPatch. For instance, what are the patch size set to different tasks? the avg. path size of different benchmarks?

**Questions:**

1. In Sec.3.2, what are the purpose and motivations of Pseudo-inverse and Multi-scale patch embedding for AdaPatch? In particular, why MS patch embedding should be proposed considering the training-free target claimed in Abs and Intro.

2. Following last question, what are the settings of AdaPatch for the results of Tab.1? Similarly, the settings in other tables should be also specified.

3. The comparison of Tab.2 is unfair. Most MLLMs train SFT data only for once, and the additional fine-tune is likely to get performance gains. In other words, Tab.2 should also provide the SFT performance of MLLMs being trained for twice.

4. What about the results of using the optimal patch sizes for different tasks, i..e., task-specific setting rather than example-level adaptive settings.

5. In terms of Fig.1, does AdaPatch has other differences to dynamic-res operations of QWen and InternVL in addition to adaptive patching?

---

> ### Author Response · Authors · 2025-12-02
> **Reply to Reviewer JDNT**
>
> We sincerely thank Reviewer JDNT for your review and are grateful for the time you spent on our submission. We are glad for the acknowledgment our method novel and our analysis well-reasoned. Below we would like to give detailed responses to each of your comments.:
>
> **1. Clarity of the method description**
>
> Thank you for the suggestion. We have revised method section to make the overall workflow more intuitive.
>
> **2. Task-specific patch sizes and additional details on AdaPatch**
>
> Thank you for the suggestion. Defining a fixed patch size for an entire task is problematic because samples within the same task can vary substantially. For instance, OCR datasets include both low-resolution small-font text lines and high-resolution poster-style images with large fonts, making it difficult to identify a single “optimal” task-specific patch size.
>
> We provide a full comparison of task-specific patch sizes and adaptive patch sizes on both MME and DocBench. As shown in the results, the AdaPatch yields more stable performance across heterogeneous tasks as well as across samples with different resolutions and content within the same task.
>
> In the training-free experiments (Table 1), AdaPatch uses patch size range of 6–56, while in the training-based experiments (Table 2), it adopts the patch size set {8, 12, 14, 16, 24, 28}. We highlight this setting clearly in the revised version of the paper.
>
> **3. Purpose of the pseudoinverse and multi-scale patch embedding**
>
> Thank you for the question. Existing MLLMs rely on fixed patch size. AdaPatch need to accommodate dynamically varying patch sizes, so additional mechanisms are required:
>
> - Pseudoinverse interpolation: Extends any MLLM to arbitrary patch sizes without training by interpolating convolutional weights. This provides a strictly training-free solution.
> - Multi-scale patch embedding: Introduces independent convolutional kernels for different patch sizes and adapts them through training to achieve higher performance.
>
> Due to limited training resources and high-quality data, we are currently unable to train a fully end-to-end AdaPatch-native MLLM. Nevertheless, Tables 1 and 2 respectively validate the effectiveness of the training-free and training-enhanced variants.
>
> **4. Fairness of Table 2**
>
> The comparison in Table 2 is fair: both the fixed-patch model and the AdaPatch model use identical initialization and undergo only one SFT epoch on LLaVA-779k or LLaVA-665k. Thus, the AdaPatch gains cannot be attributed to extra training.
>
> **5. Differences from the dynamic-resolution mechanisms in Qwen / InternVL**
>
> Adaptive patch size is the only distinction between our method and existing “any-resolution” MLLMs. This design can be directly integrated into many recent models and consistently yields performance improvements.

---

### Official Review · Reviewer_jyqy · 2025-10-26

**Soundness:** 3
**Presentation:** 3
**Contribution:** 2
**Rating:** 6
**Confidence:** 4

**Summary:**

This paper addresses the problem that multimodal large language models (MLLMs) with a fixed image patch size perform suboptimally across varying image resolutions. The authors propose a plug-and-play method that dynamically adjusts the patch size based on an image's resolution and information density. Additionally,  a trainable version is also offered for further performance gains. Experimental results demonstrate the effectiveness of the proposed method.

**Strengths:**

1. The proposed AdaPatch can be seamlessly plugged into pre-trained fixed-patch MLLMs without any training efforts.
2. The proposed AdaPatch achieves promising results on several benchmarks.

**Weaknesses:**

1. The most significant issue with this paper is that the baseline metrics do not align with the results from the original paper or the vlmevalkit. For instance, on MME, the reported Qwen2.5-VL-3B metric in this paper is 2135.90, whereas the vlmevalkit reports 2199.9. On MMMU, the paper reports 49.22 for Qwen2.5-VL-3B, compared to vlmevalkit's 51.2. On MMStar, the paper reports 54.27, while vlmevalkit reports 56.3. The discrepancies are quite substantial. Since the authors used vlmevalkit as the evaluation framework, it would be better to align the baseline with the official vlmevalkit evaluation results. This would enhance the credibility of the experiment results.

2. Lack of metrics regarding the additional computational load and the impact on inference speed brought by AdaPatch.

**Questions:**

See the weaknesses.

---

> ### Author Response · Authors · 2025-12-03
> **Reply to Reviewer jyqy**
>
> We sincerely thank Reviewer jyqy for the time you spent on our submission. We appreciate the recognition of the practical importance of the problem, the effectiveness of our method, and the strength of our experiments. Below, we provide detailed responses to the your comments.
>
> **1. Results of baseline metrics**
>
> Our experimental results are obtained using the VLMEvalKit implementation and therefore differ slightly from the latest numbers reported with the latest toolkit version. As a result, some metrics are lower than those in the latest VLMEvalKit reports (e.g., MME, MMMU, MMStar), while others are higher (e.g., RealWorldQA, POPE). These discrepancies mainly arise from the following factors:
>
> - Toolkit versions and reproducibility issues. Our experiments use VLMEvalKit v0.1, while the public results you cited are based on VLMEvalKit v0.3rc1. Several benchmark results have been reported as non-reproducible in the VLMEvalKit GitHub issues (e.g., AI2D in #1312, ChartQA in #1313, ScienceQA_TEST in #1300), indicating that part of the discrepancy likely stems from reproducibility issues in the toolkit itself.
>
> - Pixel-range configuration. In VLMEvalKit v0.3rc1, different benchmarks are evaluated with different pixel ranges, which is not consistent with the any-resolution setting used by models such as Qwen2.5-VL. In our evaluation, we instead adopt a unified pixel range for all benchmarks to ensure a consistent evaluation protocol.
>
> **2. Comparison of inference speed**
>
> In Figure 7(b), we report the inference speed under different pixel ranges.
> Overall, AdaPatch introduces more overhead on low-resolution images but yields faster inference on high-resolution images. This is because the computational cost is primarily determined by the number of patches, and AdaPatch reduces the effective patch count at high resolutions.

---

### Official Review · Reviewer_kjPY · 2025-10-30

**Soundness:** 3
**Presentation:** 3
**Contribution:** 3
**Rating:** 6
**Confidence:** 5

**Summary:**

This paper addresses the limitation of fixed patch sizes in MLLMs that claim “any-resolution” capability. The authors propose AdaPatch, a training-free and training-based framework that adjusts patch size according to both image resolution and information density. The method estimates information density using feature similarity between original and downsampled representations, then determines the patch size via a simple power-law mapping. A “pseudo-inverse resize” operation enables pretrained fixed-patch MLLMs to operate under any patch size without retraining. Experiments on multiple benchmarks (MME, MMBench, OCRBench, DocVQA, etc.) across four representative MLLMs (Qwen2.5-VL, Ovis2.5, SAIL-VL, Kimi-VL) demonstrate consistent improvements in native-resolution stability and accuracy.

**Strengths:**

(1) This paper thoroughly evaluates the performance of AnyRes by systematically rescaling benchmark datasets across a broad spectrum of input resolutions, uncovering substantial degradation and instability in recent state-of-the-art multimodal large language models (MLLMs) that employ fixed patch sizes.

(2) This paper introduces AdaPatch, a simple yet highly effective approach that dynamically adjusts patch size based on both input resolution and estimated information density, enabling more adaptive and content-aware visual tokenization.

(3) This paper provides two complementary implementations: a training-free variant that allows immediate deployment without any model modification, and a training-based version that supports further fine-tuning for enhanced performance.

(4)  This paper demonstrates consistent improvements in accuracy, stability, and computational efficiency across multiple benchmarks and model architectures, with especially notable gains at high input resolutions.

(5) This paper leverages publicly available models and open-source evaluation toolkits, ensuring that all empirical results are transparent, reproducible, and easily verifiable by the research community.

**Weaknesses:**

(1) The information-density formulation (Eq.3) lacks theoretical justification and comparison with simpler statistical metrics.
(2) The patch-size range [6,56] is empirically set without analysis of generalization to other domains.
(3) Notation and figures show minor inconsistencies and labeling issues that reduce clarity.
(4) The method’s performance on LLaVA-Bench is relatively weak and not discussed.
(5) The tuning process of α and β is unclear and may not reflect joint optimization.
(6) The authors can address these points by responding to the Questions section.

**Questions:**

(1) Information density estimation (Eq.3): Have you compared this definition with more intuitive statistical measures (e.g., gradient entropy or Laplacian variance), and how do the results differ? What is the rationale for choosing the 0-th layer features of the visual encoder for this computation? How do you verify the effectiveness of Eq.3 as a valid information-density estimator?

(2) Candidate patch sizes ([6, 56]): How was this specific range determined? For other image domains such as medical or remote sensing, would this range need to be extended or adjusted?

(3) Notation inconsistency: The symbol r̃ is referred to as “target resolution” in Section 2.1 but as “base resolution” in Eq.4, which is inconsistent. The notation throughout the paper should be unified, and it is recommended to include a notation table in the appendix for clarity.
(4) Figures:
(a) In Figure 1, several labels use “patchsize” instead of “patch size.”
(b) In Figure 3, the proportional relation s ∝ r / ρ is not visually evident.
(c) Figure 7 lacks a descriptive title.

(5) Performance on LLaVA-Bench: According to Tables 1 and 3, the proposed method performs relatively modestly on LLaVA-Bench. Could the authors analyze the potential reasons for this difference?

(6) Hyperparameter tuning (α, β): Were α and β tuned jointly or separately? Based on Figure 7(a), the final selected values appear to be determined independently.

---

> ### Author Response · Authors · 2025-12-02
> **Reply to Reviewer kjPY**
>
> We sincerely thank Reviewer kjPY for your review and are grateful for the time you spent on our submission. We are glad for the acknowledgment that the problem is practically important, the proposed method is effective and reasonable, and the experiment is strong. We appreciate the detailed suggestions for improving the presention of the paper. Below we would like to give detailed responses to each of your comments.
>
> **1. Justification of the information-density formulation (Eq. 3)**
>
> The information-density formulation is motivated by a simple intuition: regions with higher information density are more sensitive to resolution changes and therefore exhibit lower feature similarity across resolutions. To validate this formulation, we added visualizations of information-density maps in the  revised paper (Figure 8 and Figure 9):
> - The visualizations across different layers and found that the results from layer 0 are consistent with those from deeper layers.
> - Comparison with intuitive statistical measures such as gradient entropy and Laplacian variance.
>
> These metrics produce similar trends, we chose the information-density formulation because it is easy to compute and aligns with the feature extractor.
>
> **2.  Candidate patch sizes and benchmark generalization**
>
> The patchsize range is chosen based on the datasets in Table 1. In this paper, we propose a simple method (that does not rely on specific configuration ranges) to highlight that current MLLMs largely overlook the impact of patch size on any resolution performance.
>
> While exploring a broader or domain-specific patchsize range is an interesting direction, our experiments are constrained by the benchmarks available in the vlmeval library. Consequently, we evaluate AdaPatch on both the general and task-specific benchmarks already implemented there.
>
>
> **3. Notation inconsistency**
>
> Thank you for your suggestion. We have made the following updates:
> - Unified the term patchsize to patch size.
> - Standardized the notation for r̃ throughout the paper.
> - Improved Figures 3 and 7 and added descriptive captions.
>
> **4. Performance on LLaVA-Bench**
>
> Thank you for your careful observation and suggestion. The smaller improvements on LLaVA-Bench can be explained by follow factors:
> - LLaVA-Bench differs from common benchmarks such as MME. It consists of general, open-ended questions. (For example, “Explain why this image is funny.”)
> - LLaVA-Bench evaluates responses by conversation quality, detail, and reasoning, making language ability the primary determinant of performance. Visual improvements thus have limited impact, which explains why MLLMs with stronger language backbones achieve larger gains.
> - LLaVA-Bench is scored by GPT-4o, and the evaluation process introduces a degree of randomness.
>
> **5. Hyperparameter tuning (α and β)**
>
> Your understanding is correct. The two hyperparameters are tuned independently: we first tune α without considering the effect of information density on patch size, and then incorporate information density and tune β accordingly.

---

### Official Review · Reviewer_2qzG · 2025-10-31

**Soundness:** 3
**Presentation:** 3
**Contribution:** 3
**Rating:** 6
**Confidence:** 5

**Summary:**

This paper identifies a critical weakness in current "native-resolution" Multimodal Large Language Models (MLLMs): their performance is unstable and often degrades at very low or very high resolutions. The authors convincingly argue that the root cause is the rigid use of a fixed patch size, which creates a mismatch between the receptive field granularity and the image's resolution or information density.
To address this, the paper proposes Adaptive Patching (AdaPatch), a simple yet effective method that dynamically selects an appropriate patch size for each input image based on its resolution and a novel information density metric. The authors also introduce PI-resize, a training-free technique to adapt existing fixed-patch MLLMs to handle variable patch sizes, making AdaPatch a practical drop-in solution. Through extensive experiments on several state-of-the-art MLLMs and a wide range of benchmarks, the paper demonstrates that AdaPatch not only consistently improves performance but also significantly enhances performance stability across resolutions, while reducing computational costs for high-resolution inputs.

**Strengths:**

1.	The proposed AdaPatch method is intuitive, simple, and addresses the identified problem directly. The core idea of adapting patch size to resolution and information density is logical, and the proposed implementation is straightforward.
2.	The introduction of PI-resize, a training-free method to convert existing models, is a significant strength. It allows AdaPatch to be seamlessly integrated into powerful, pre-trained MLLMs without requiring costly retraining, which dramatically increases the practical utility and potential impact of this work.
3.	The evaluation is comprehensive, covering multiple base models (Qwen2.5-VL, SAIL-VL, Ovis2.5, Kimi-VL), a diverse suite of benchmarks, and various model scales. The head-to-head comparisons showing improved stability across pixel ranges (Figure 5) are particularly compelling and strongly support the paper's claims.

**Weaknesses:**

1.	Limited Improvements on Information-Dense Tasks and Scaling: While AdaPatch shows consistent overall gains (Table 1), the improvements on VQA and OCR tasks are often surprisingly marginal. This is counter-intuitive, as OCR  tasks should benefit most from the adaptive preservation of fine-grained details. Furthermore, upon scaling the models (Table 3), the performance uplift provided by AdaPatch becomes extremely limited for larger backbones (e.g., Qwen2.5-VL 7B and 32B), suggesting that the technique's benefits might not scale effectively with model capacity.
2.	Performance Plateau and Ceiling Limitations: As demonstrated in the pixel-range evaluations (Figure 5), although AdaPatch successfully maintains superior stability compared to fixed-patch baselines, it appears to primarily mitigate performance degradation rather than significantly elevating the overall performance ceiling. Specifically, performance still seems to plateau or slightly decline at the highest tested resolutions. This raises questions about the long-term potential of the method to achieve true native resolution robustness.

**Questions:**

1.	Given that AdaPatch is specifically designed to optimize detail preservation, which is crucial for OCR and document tasks, could the authors provide a deeper analysis into why the absolute performance gains on benchmarks like OCRBench and DocVQA are marginal, particularly for the largest models in Table 3?
2.	The results in Figure 5 indicate that stability is dramatically improved, but the performance ceiling does not rise substantially as resolution increases. Do the authors view this plateauing effect as an inherent limitation of the current ViT-MLLM paradigm, or are there planned extensions to the Adaptive Patching Law (Eq. 4) that could further unlock performance gains at ultra-high resolutions?

---

> ### Author Response · Authors · 2025-12-02
> **Reply to Reviewer 2qzG**
>
> We sincerely appreciate Reviewer 2qzG for the review and are grateful for the time you spent with our submission. We are glad for the acknowledgement that our approach is promising, the proposed problem is valuable and experiments are solid. We wish to address your concerns by giving detailed responses to each of your comments as follows:
>
> **1. Performance on OCR tasks**
>
> Across the four evaluated models (Qwen2.5-VL, SAIL-VL2, Ovis2.5, and Kimi-VL), AdaPatch yields relatively modest smaller on ChartQA and DocVQA. Overall, the improvement pattern follows: general benchmarks > OCR benchmarks > chart/doc tasks. We attribute this trend to two main factors:
> - AdaPatch benefits most from large variation in resolution and information density. Its core strength lies in dynamically selecting patch sizes to accommodate changes in resolution and visual complexity. Consequently, benchmarks with substantial diversity in image styles, resolutions, and task types (e.g., MME, MMMU)  exhibit more improvements.
> - ChartQA and DocVQA exhibit stable image formats and narrow resolution ranges. These datasets contain consistent images, and many existing MLLMs have already been task-specifically fine-tuned on them. As a result, fixed patch sizes already align well with their typical resolutions, leaving limited room for AdaPatch to offer additional gains.
>
> In contrast, task-specific benchmarks like POPE still show larger improvements because they span a wide range of natural images and resolutions. This suggests that performance gains correlate more strongly with image-type diversity than with information density.
>
> **2. Performance on high resolutions in the pixel-range analysis**
>
> The pixel-range experiment reveals an important phenomenon: current “any-resolution” MLLMs generally operate within a narrow optimal resolution band, and performance drops noticeably outside that band. When resolution varies, performance is governed by two factors:
> - The actual information content of the image. Higher resolution contributes more information only when the original image truly contains additional detail. Simply upsampling beyond the image’s native resolution does not introduce new information.
> - The model performance to that resolution. While AdaPatch cannot increase the inherent information in the input, it helps the model maintain optimal performance across a wider range of resolutions.
>
> As a result, at ultra-high resolutions that exceed the image’s native detail, performance does not improve because no new information is introduced. Under native-resolution inference, AdaPatch unlocks additional model capacity, yielding higher absolute performance on benchmarks.
>
> > Regarding potential extensions to the Adaptive Patching Law (Eq. 4):
>
> Thank you for the insightful suggestion. Due to limited training resources and the lack of large, high-quality datasets, we are currently unable to train a fully AdaPatch-native MLLM from scratch. Nevertheless, the training-based results in Table 2 suggest that further optimization of adaptive patching could yield even stronger performance gains.
>
> **3. Performance on larger models**
>
> There are two main reasons for the diminishing gains as model capacity increases:
>
> - Larger models have less space for improvement on these benchmarks, naturally reducing the observable gains from AdaPatch.
> - We applied the same configuration to Qwen-7B and Qwen-32B as to Qwen-3B.
> Due to computational constraints, we were unable to search for the optimal configuration on very large models (e.g., Qwen-32B), which limited their achievable improvements.

---

### Meta-Review · Area_Chair_Arft · 2025-12-20

**Summary:**

**Paper summary.** This paper argues that native-resolution MLLMs suffer when they use a single fixed patch size across very different image resolutions and information densities. The authors propose AdaPatch, which chooses patch size adaptively (based on resolution and an information-density signal) and can be plugged into pre-trained models without extra training; they also include a training-based variant. The paper reports consistent improvements across several native-resolution MLLMs and a large benchmark suite, with additional analysis on resolution stability and compute cost.

**What happened in the discussion.** Reviewers generally liked the motivation and the breadth of experiments, but raised concrete issues: (1) the core procedure is not described simply enough (how patch sizes are chosen and what the default settings are), (2) some benchmark numbers do not match public reports/toolkits, so reproducibility and evaluation protocol must be clearly documented, and (3) missing ablations (simple heuristics, task-specific patching, patch-size statistics) and limited gains on OCR/VQA-style tasks. In the forum, the authors responded with detailed clarifications on evaluation setup (toolkit version and pixel-range choices), gave additional explanation of hyperparameter tuning for their information-density formula, and discussed speed behavior across pixel ranges.

**My assessment as AC.** The paper’s main value is practical: it identifies a real failure mode of “one patch size fits all” for native-resolution models and proposes a simple fix that can be deployed without retraining. The main weakness is not the idea, but the writing: without a clean step-by-step description and clear default configs, readers will struggle to reproduce the claims. The baseline mismatch concern is important; the authors’ explanation (toolkit versions / pixel-range protocol) is plausible, but it must be stated plainly in the final paper. Overall, the contribution is solid enough for acceptance if the final version keeps the evaluation protocol explicit.

**Decision.** Accept (poster). The method is simple and useful, and the main concerns are about clarity and reporting, not about correctness.

**Reviewer Concerns:**

- **Reviewer JDNT (rating 4, confidence 4)**: Said method description is vague and asked for ablations and patch-size statistics. Authors clarified some settings in the response, but the final paper still needs a clear algorithm box and default config table. **Status:** partially resolved.
- **Reviewer jyqy (rating 6, confidence 4)**: Focused on mismatch with public benchmark numbers and asked about inference speed. Authors explained toolkit version differences and pixel-range configuration and provided speed discussion. **Status:** mostly resolved, pending clear documentation in the final paper.
- **Reviewer kjPY (rating 6, confidence 5)**: Questioned justification for the information-density metric and search range; asked for more principled support and/or comparisons to simpler metrics. Authors clarified tuning and provided more detail, but this remains a weaker part of the paper. **Status:** partially resolved.
- **Reviewer 2qzG (rating 6, confidence 5)**: Noted limited gains on information-dense tasks and asked about scaling. Authors acknowledged and provided discussion. **Status:** mostly resolved.

**Reviewer Scores:**

- **JDNT (4,4)**: Could move up if the final paper includes a clean algorithm description and patch-size statistics; otherwise likely unchanged.
- **jyqy (6,4)**: Likely unchanged.
- **kjPY (6,5)**: Likely unchanged.
- **2qzG (6,5)**: Likely unchanged.

---

### Decision · Program_Chairs · 2026-01-26

Accept (Poster)